# Identification of Polyphenol Derivatives as Novel SARS-CoV-2 and DENV Non-Nucleoside RdRp Inhibitors

**DOI:** 10.3390/molecules28010160

**Published:** 2022-12-25

**Authors:** Shenghua Gao, Letian Song, Hongtao Xu, Antonios Fikatas, Merel Oeyen, Steven De Jonghe, Fabao Zhao, Lanlan Jing, Dirk Jochmans, Laura Vangeel, Yusen Cheng, Dongwei Kang, Johan Neyts, Piet Herdewijn, Dominique Schols, Peng Zhan, Xinyong Liu

**Affiliations:** 1Department of Medicinal Chemistry, Key Laboratory of Chemical Biology (Ministry of Education), School of Pharmaceutical Sciences, Cheeloo College of Medicine, Shandong University, Jinan 250012, China; 2Department of Pharmacology, School of Pharmaceutical Sciences, Cheeloo College of Medicine, Shandong University, Jinan 250012, China; 3Shenzhen Research Institute of Shandong University, A301 Virtual University Park in South District of Shenzhen, Shenzhen 518057, China; 4Laboratory of Virology and Chemotherapy, Rega Institute for Medical Research, Department of Microbiology, Immunology and Transplantation, KU Leuven, 3000 Leuven, Belgium; 5Laboratory of Medicinal Chemistry, Rega Institute for Medical Research, KU Leuven, 3000 Leuven, Belgium

**Keywords:** SARS-CoV-2, DENV, polyphenol, RdRp, non-nucleoside

## Abstract

The Coronavirus Disease 2019 (COVID-19) and dengue fever (DF) pandemics both remain to be significant public health concerns in the foreseeable future. Anti-SARS-CoV-2 drugs and vaccines are both indispensable to eliminate the epidemic situation. Here, two piperazine-based polyphenol derivatives **DF-47** and **DF-51** were identified as potential inhibitors directly blocking the active site of SARS-CoV-2 and DENV RdRp. Data through RdRp inhibition screening of an in-house library and in vitro antiviral study selected **DF-47** and **DF-51** as effective inhibitors of SARS-CoV-2/DENV polymerase. Moreover, in silico simulation revealed stable binding modes between the **DF-47**/**DF-51** and SARS-CoV-2/DENV RdRp, respectively, including chelating with Mg^2+^ near polymerase active site. This work discovered the inhibitory effect of two polyphenols on distinct viral RdRp, which are expected to be developed into broad-spectrum, non-nucleoside RdRp inhibitors with new scaffold.

## 1. Introduction

Coronavirus and flavivirus both encode single-stranded positive sense RNA (+RNA), containing some viruses seriously threatening human health. Severe Acute Respiratory Syndrome Coronavirus 2 (SARS-CoV-2) of *β*-coronavirus, as the causative pathogen of Coronavirus Disease 2019 (COVID-19), is now still spreading worldwide [1]. The clinical manifestations of COVID-19 vary from asymptomatic to common fever, and up to more severe symptoms such as pneumonia, respiratory failure, multiorgan failure, and eventually death [2]. According to the World Health Organization (WHO), there were over 630 million confirmed infections and 6.6 million deaths globally in mid-November 2022 [3]. The emergence of the Omicron variant, which is highly infectious and resistant to the vaccines, has reduced the effectiveness of the vaccine by at least 40% [4]. Dengue virus (DENV) belongs to a serotype subgroup of flavivirus which is mainly insect-borne, causing dengue fever (DF), dengue hemorrhagic fever (DHF), and dengue shock syndrome, with high incidence rate and mortality. Currently, in some underdeveloped countries and areas, the regular outbreaks of dengue fever form a double threat along with COVID-19 pandemic [5,6]. Among these territories, the DENV-SARS-CoV-2 co-infection has been a serious situation, which has been one of the focus issues on COVID-19 epidemiology. Thus, developing broad-spectrum antiviral agents blocking both SARS-CoV-2 and DENV is a promising strategy to alleviate the public health burden in certain areas around the world [7,8]. Although supportive therapy of dengue fever remarkably reduced the mortality rate, no specific therapy agents towards DENV were approved. To end the global pandemic and prevent future outbreaks of highly contagious RNA viruses, antivirals are expected to act as essential complements to vaccines.

RNA-dependent RNA-polymerase (RdRp) acts as a critical component in the life cycle of both SARS-CoV-2 and DENV. In the two viruses, RdRp adapts the “Finger-Palm-Thumb” conformation formed by several conserved motifs, and utilizes two sequential aspartate residues coordinated with Mg^2+^ ions as their catalytic centers [9,10,11,12,13]. The conservation of RdRp among evolutionary distant RNA viruses and the absence of host homologs apparently make it an ideal target for the development of potential antivirus drug candidates to multiple viruses.

RdRp inhibitors can be classified into nucleoside inhibitors (Nis) and non-nucleoside inhibitors (NNIs), based on their structures. Nucleoside inhibitors bind to RdRp active center and could be incorporated into elongating RNA chain, causing chain termination or lethal mutagenesis. Four repurposed SARS-CoV-2 RdRp inhibitors: Remdesivir, favipiravir, molnupiravir, and bemnifosbuvir (AT-527) were investigated for their anti-COVID-19 activity (Figure 1) [14]. Among them, remdesivir (EC_50_ = 0.77 μM) was the first drug urgently approved to treat COVID-19 in 2020 by the FDA. However, several clinical trials have demonstrated that the time point of efficacy of remdesivir was not significantly earlier than that of placebo group [15], and it has failed to significantly improve mortality, cure rate, recovery time, and other indicators [16,17,18]. Favipiravir, as a pyrazinecarboxamide derivative, requires in vivo conversion to ribosylated triphosphate favipiravir, to act as an antiviral agent. The complex structure of favipiravir and SARS-CoV-2 RdRp was revealed by electron cryomicroscopy, demonstrated the unexpected base pairing pattern between favipiravir and pyrimidine residues, and explained its ability to mimic adenine and guanine nucleotides [19]. Early studies showed favipiravir could alleviate symptoms and shorten hospitalization; thus, it has been approved in India and Russia. However, FUJIFILM Toyama Chemical Co., Ltd. announced the cessation of the development of the anti-influenza drug Avigan for COVID-19 infection on October 14, 2022 [20]. Molnupiravir (EC_50_ = 0.3 µM) induces lethal mutagen in SARS-CoV-2 genome. By the end of 2021, FDA issued emergency use authorization for molnupiravir for it led to a 50% decrease of hospitalization or death in a clinical trial, but only for non-hospitalized patients when no other therapies are available [21]. Bemnifosbuvir (EC_90_ = 0.47 µM) is a repurposed SARS-CoV-2 RdRp inhibitor, but failed in phase II clinical trial of treating COVID-19 [22]. Further phase II and III trials are setting out to evaluate whether bemnifosbuvir could be applied in the future [23]. Meanwhile, bemnifosbuvir was also reported to effectively inhibit DENV2/DENV3 in vitro (EC_50_ = 0.48/0.77 µM) and in vivo [24]. The current situation urgently calls for more RdRp inhibitors with definite therapeutic effect. 

Non-nucleoside inhibitors may act directly on RdRp active site or bind to its potential allosteric sites, thus impairing polymerase function [14]. Compared with nucleoside inhibitors, NNIs do not incorporate into RNA chains, therefore can evade the exonuclease (Nsp14) excision mechanism, and have greater potential of modification and development. Meanwhile, targeting the catalytic metal ions by chelators has been proved successful in inhibiting HIV, HCV, and influenza virus [25,26,27]. The strategy of utilizing catalytic Mg^2+^ of SARS-CoV-2/DENV RdRp as an anchoring point of metal chelating agents had been reported. Flavonoids with polyphenol moiety display potent inhibition towards SARS-CoV-2 RdRp in computational simulation or experiments. Cyanidin 3-O-rutinoside and petunidin 3′5-O-diglucoside exhibited outstanding binding affinity in virtual screening [28], while Baicalein (EC_50_ = 4.5 μM) and Baicalin (EC_50_ = 9.0 μM) are potent SARS-CoV-2 inhibitors in both RdRp enzymatic assay and cellular assay (Figure 2) [29]. Docking results highlighted that the pyrogallol group in these compounds provides multiple hydrogen bond interactions or chelates with Mg^2+^ in the RdRp active site. Pyridoxine derivative DMB220 (EC_50_ = 2.7 ± 0.6 μM) and one quinolone-like compound (EC_50_ = 3.3 ± 0.5 μM) with metal-chelating groups [30,31], are reported to inhibit DENV. Yet no metal chelators targeting virus RdRp have been approved for antivirus therapy.

In this study, in order to search for potent inhibitors of SARS-CoV-2 and DENV RdRp, we screened metal ion chelators from an in-house compound library, based on our SARS-CoV-2 RdRp inhibition screening assay. Two compounds **DF-47** and **DF-51** were chosen, then re-evaluated for their IC_50_ values and cellular activities in vitro. Docking studies and molecular dynamic simulation studies identified them as active inhibitors towards both SARS-CoV-2 and DENV RdRp. Discovery of these novel inhibitors offers future choice of SARS-CoV-2 inhibitor development, and reveals critical residues around RdRp catalytic site for target-based drug designing.

## 2. Results and Discussion

### 2.1. In-House Library Screening For Potent Rdrp Inhibitors

In this study, we evaluated RdRp binding activity of an in-house compound library containing metal ion chelators with piperazine scaffold (over 100 compounds of polyphenol, 1-hydroxy-1,8-naphthyridinone, 5-hydroxypyrido [2,3-b]pyrazinone, and 4-hydroxyquinazolinone analogues, originally designed and synthesized for anti-HIV-1, see Appendix A) [32,33,34,35,36] screened by established high-throughput enzyme assays of RdRp complex, as novel antiviral therapeutic candidates for COVID-19 through target activity verification and cell-based activity evaluation. A polymerase reaction system was built to evaluate the residual activity of RdRp in the presence of inhibitors [37,38]. Recombinant non-structural proteins were expressed, purified, and mixed in assay buffer to form the SARS-CoV-2 polymerase complex. The RNA template applied in the assay was specially designed to form a hairpin structure at 3’ end as the primer of polymerase reaction, and carries a 6-FAM label at 5’ end. After a reasonable time for polymerase reaction progress, the system was immediately quenched. The produced RNA chain was visualized and quantified on gel. The amount of full-length RNA product relates to RdRp activity. Preliminary screening involves all compounds under a single concentration of 50 μM. A total of 11 compounds with top inhibition rates were selected for RdRp inhibition tests under gradient concentrations (listed in Table 1). The visualized results from the gel experiments are displayed in Figure 3.

### 2.2. Evaluation of Virus Inhibition Activity 

For the selected compounds listed above, we further evaluated their enzymatic inhibition potency under three concentration gradients (5, 10, and 20 μM), and the density of visualized stripes represent concentrations of RNA template and product in each well. At a concentration of 20 µM, most of the compounds still exhibited significant inhibition of RdRp activity. At 10 μM, **DF-36**, **DF-47**, **DF-51**, and Baicalein remained partially active, whereas at 5 µM, only limited activity was observed (Figure 3c). To obtain accurate IC_50_ values for **DF-47** and **DF-51**, both compounds were again tested under seven concentrations ranging from 0.4 to 100 μM (Figure 3d) to depict full dose–response curves. **DF-47** turned out to be the most potent SARS-CoV-2 RdRp inhibitor with an IC_50_ of 9.2 ± 1.1 μM. Unlike RDV-TP which causes delayed chain-termination during RNA synthesis via incorporation of the nucleotide analog by the RdRp [39,40], these polyphenols are most likely to act by chelating with the magnesium ions of active center, thus directly blocking SARS-CoV-2 RdRp function. 

To further exploit pan-RNA virus inhibition of the compounds in Table 1, the compounds were evaluated as potential inhibitors of the RdRp domain of DENV NS5. We used a fluorescent-based assay to screen the compounds from Table 1 as potential DENV RdRp inhibitors. Only **DF-47** and **DF-51** were endowed with detectable inhibition towards DENV RdRp (Table 2). **DF-51** displayed a low-micromolar IC_50_ value of 4.8 ± 0.7 µM towards DENV RdRp, whereas **DF-47** had relatively weaker inhibition, in contrast to the results of SARS-CoV-2 RdRp.

### 2.3. Antiviral Activity of DF-47 and DF-51

Subsequently, we measured the antiviral activity of **DF-47** and **DF-51** towards SARS-CoV-2 and DENV3 in the BSL-3 (biosafety level 3) laboratory, along with their cytotoxicity to corresponding cell lines. Remdesivir and Baicalein were employed as positive controls in cellular SARS-CoV-2 inhibition tests. The test results were depicted in Table 3. The activity of the positive control was consistent with that reported in the literature [29], which proved the validity of our method. Unfortunately, **DF-47** and **DF-51**, despite their good inhibitory activity against RdRp, displayed no significant antiviral activity against SARS-CoV-2 in vitro. However, both compounds showed moderate inhibition of DENV3 in A549 cells with measurable EC_50_ values, consistent to their IC_50_ values. In addition, **DF-47** and **DF-51** displayed no cytotoxicity for either Vero E6 or A549 cells (CC_50_ > 100 μM).

The absence of antiviral activity towards SARS-CoV-2 might be due to the poor membrane permeability resulted from multiple phenolic hydroxy groups. To verify this, we utilized the Membrane permeability prediction module [41,42] in Schrödinger suites to quantitate cell-entering property of **DF-47** and its analogs, along with two proposed prodrugs **DF-47**-pro/**DF-51**-pro (tri-isobutyryl ester). The permeability was evaluated by the Membrane dG Insert, of which higher absolute value represents lower permeability. As shown in Table 4, all the DF series compounds possessed extraordinarily low predicted permeability, compared with their prodrugs. This explains why **DF-47** lacked cellular activity and hints for further modifications. Therefore, we will further chemically modify the compound according to the design strategy of prodrug.

### 2.4. In Silico Study 

As **DF-47** and **DF-51** are non-nucleoside inhibitors with polyphenol group, their possible mechanism of inhibition might be through binding to the RdRp by chelating with the Mg^2+^ at the active site. In order to explore their mechanism, we conducted in silico studies to determine the possible binding mode between **DF-47** and SARS-CoV-2 RdRp. 

First, docking results demonstrated that **DF-47** could bind to SARS-CoV-2 RdRp with a binding energy of −7.916 kcal/mol, which displayed stronger binding energy than that of Remdesivir (−6.5 kcal/mol) [40]. Most importantly, **DF-47** is likely to chelate with two Mg^2+^ of RdRp. Therefore, it is a reasonable assumption that **DF-47** inhibits RdRp activity through chelating with Mg^2+^ and blocking its active site.

Figure 4a,b shows the 2D-interaction diagram and the 3D docking pose of **DF-47**. The two Mg^2+^ directly interacted with the phenolic hydroxyl groups, enabling **DF-47** to anchor in the entry tunnel of NTPs. Meanwhile, Tyr619, Asp760, Asp761, and Glu811 bind to the ligand through metal coordination bonds. Other interactions with essential residues in conserved motifs or with amino acids responsible for NTP recognition are also observed [37]; for example, the hydrogen bond interaction between Cys799 and the amino group of **DF-47**, as well as between Lys551 and the amide oxygen in the ligand structure. The benzenesulfonamide and bisphenyl moieties have Pi-Pi stacking with His810 and His439, respectively, whereas Arg836 and phenyl generated Pi-cation interaction. These interactions with free RdRp may lead to a loss in its recognition and catalytic activity. For DF-51, it adopted a slightly different binding position, yet still maintained metal coordination with the two magnesium ions (Figure 4c,d), according to the docking results.

The docking study also revealed multiple interactions between **DF-51** and RdRp domain of DENV3 NS5 (Figure 5 a,b). The binding energy reached −8.061 kcal/mol, while chelation of the single Mg^2+^ and several hydrogen bonds were hypothetically established. In addition to binding with metal ion, the polyphenol moiety formed cation-π stacking with protonated Lys689 by its electron-rich phenyl ring. Two amide oxygen atoms snugly contacted with Cys709 and Ser710, together with the cyano group reaching backbone NH of Gly536. These H-bonds are supposed to stabilize **DF-51** at the hydrophilic pocket of RdRp catalytic center. At the far end, the biphenyl sidearm fitted into a shallow groove formed by Tyr606 and Ile797, which accounts for additional hydrophobic contacts. This bulky group is supposed to restrict the conformational changes of priming loop, which is an essential moiety in catalytic function [12,13]. Given that the Mg^2+^ of DENV3 RdRp may not mediate catalytic process [43], inhibition of **DF-51** most likely resulted from blocking the NTP tunnel and disturbing priming loop conformation. In another docking study, DF-47 exhibited a similar binding mode with that of DF-57, but formed less interactions with nearby residues (Figure 5 c,d).

To further evaluate the binding stability of **DF-47** and **DF-51** towards SARS-CoV-2/DENV3 RdRp, RMSD and total energy (MMGBSA) were predicted through MD simulation systems. The average position fluctuations of ligand and protein atoms, represented by RMSD value, were shown in Figure 6. For **DF-47** complexed with SARS-CoV-2 RdRp, the ligand RMSD remained lower than that of protein RMSD, indicating its stable binding in active site. However, it is also observable that the RMSD value of **DF-47** is slightly unstable (Figure 6a). In the case of DENV3 RdRp and **DF-51**, significant fluctuations of ligand RMSD ranges from 0 to 300 ns of simulation. However, the RMSD value of **DF-51** remained stable around 2–2.5 angstroms during the last 200 ns of MD simulation, signifying its well-fitting into the active site (Figure 6b). The average ligand binding free energy of **DF-47** and **DF-51** were calculated by MMGBSA evaluation. **DF-47** has a biding energy of −124.02 kcal/mol towards SARS-CoV-2 RdRp, while **DF-51** has higher binding energy (−81.77 kcal/mol) with DENV3 RdRp, corresponding to its lower binding stability.

To evaluate the stability of chelation interaction between pyrogallol group and Mg^2+^, the distance between 4-OH group and Mg^2+^ of SARS-CoV-2/DENV3 RdRp was calculated. Both models showed a stable and close contact of phenol-OH group and Mg^2+^ in protein model during the whole process of simulation, demonstrating that their tight binding are contributed by metal chelation (Figure 7a,b).

Accumulating studies are proving the inhibition activity of polyphenols towards different enveloped RNA viruses, such as influenza, dengue, HIV, SARS-CoV, and SARS-CoV-2 [44,45]. Here, we demonstrated the antiviral activity of **DF-47** and **DF-51** against SARS-CoV-2 and DENV3 through interacting with RdRp by metal chelation. Such inhibition across different RNA viruses is noteworthy. We presume this class of compounds obtain strong interaction with RdRp by anchoring into the metal-containing catalytic center via the pyrogallol moiety, and adapting to nearby tunnels with flexible piperazine scaffold joint with H-bond donor/acceptors. However, additional research on structural biology is still needed to depict a comprehensive mechanism and key pharmacophores.

## 3. Materials and Methods

### 3.1. Compounds and the Stock Solution 

All the compounds were synthesized and published by our laboratory as racemic mixtures [32,33,34,35,36]. Each compound was prepared as a 10 mM stock solution with dimethyl sulfoxide (DMSO, Sigma Aldrich, Belgium) and then stored at −20 °C. In the cellular antivirus experiment, MEM (Gibco, NY, USA), containing 2% fetal bovine serum (FBS) was used to dilute the stock solution into gradients. 

### 3.2. Gel-based SARS-CoV-2 RdRp Assay 

Enzyme assays were performed with purified recombinant SARS-CoV-2 RdRp complex nsp12/nsp8/nsp7. The RNA sequence used for the RdRp assay is/56-FAM/rUrUrU rUrCrA rUrGrC rUrArC rGrCrG rUrArG rUrUr UrUrC rUrArC rGrCrG with hairpin structure [37,38]. RNA was annealed in 50 mM NaCl and 10 mM Na-HEPES pH 7.5 by heating the solution to 75 °C and gradually cooling to 4 °C. Reactions were carried out at 30 °C with 500 nM nsp12, 1 μM nsp7, 1.5 μM nsp8, and 200 nM RNA in the reaction buffer (20 mM HEPES, pH 7.5, 15 mM NaCl, 5% glycerol, 2 mM MnCl_2_, 1 mM MgCl_2_). Test compounds of desired concentration were added, incubated for 5 min, and the reactions were initiated by adding rNTPs to 5 μM. RNA extension reactions were stopped at the desired times by adding 2× stop buffer (8 M urea, 20 mM EDTA, 1× Tris-borate-EDTA (TBE), 0.2% Orange G). Samples were heated for 5 min at 95 °C and separated by electrophoresis in denaturing 20% acrylamide (19:1) gels (8 M urea, 1 × TBE) using BioRad Mini-PROTEAN Tetra System. The RNA products were visualized and quantified using Typhoon FLA9500 (GE Healthcare) and ImageQuant software.  Dose–response data were analyzed by nonlinear regression using GraphPad Prism 9.2.0 (Graphpad company) software. The mean of IC_50_ values and standard deviation (SD) were determined from the results of two independent experiments.

### 3.3. Fluorescent Plate DENV RdRp Assay

Expression and purification of recombinant Dengue virus NS5 has been described previously [46]. Florescent plate assay based on poly rC RNA and the fluorescent dye PicoGreen was well established for screening inhibitors against Dengue virus RdRp activity [47,48]. RdRp assays were performed in a 60 μL reaction mixture containing 1.5 μM DENV NS5, 1 µg poly rC, 100 μM GTP, various concentrations of compounds, 5% DMSO, 40 mM Tris–HCl (pH 7.0), 2 mM MnCl_2_, and 5 mM DTT. The reaction was incubated at 30 °C for 60 min and terminated with 10 mM EDTA after which 100 μL of 200-fold diluted fluorescent dye PicoGreen in TE buffer was added to each well and incubated for 5 min at room temperature. Microplate reader was used to quantitate the amount of dsRNA Fluorescence, under excitation and emission wavelengths of 480 and 520 nm, respectively. Dose-response data were analyzed by nonlinear regression using GraphPad Prism 9.2.0 software. The mean of IC_50_ values and standard deviation (SD) were determined from the results of two independent experiments.

### 3.4. Cells and Viruses

The SARS-CoV-2 isolate used in this study was the Beta-Cov/Belgium/GHB-03021/2020 (EPI ISL407976|2020-02-03). The isolate was passaged 7 times on Vero E6 cells which introduced two series of amino acid deletions in the spike protein [49]. The infectious content of the virus stock was determined by titration on Vero E6 cells. SARS-CoV-2 was used at 0.001 TCID_50_/cell.

Vero E6 cells were maintained in Dulbecco’s modified Eagle’s medium (DMEM; Gibco cat no 41965-039), supplemented with heat-inactivated 10% *v/v* fetal calf serum (FCS; HyClone) and 500 µg/mL Geneticin (Gibco cat no 10131-0275) and kept under 5% CO_2_ at 37 °C. All SARS-CoV-2-related experimental work was performed in the certified, high-containment biosafety level-3 facilities of the Rega Institute at the KU Leuven.

### 3.5. In Vitro Antiviral Assays

The SARS-CoV-2 antiviral assay is derived from the previously established SARS-CoV assay [50]. In this assay, fluorescence of VeroE6-eGFP cells (provided by Dr. K. Andries J&JPRD; Beerse, Belgium) declines after infection with SARS-CoV-2 due to the cytopathogenic effect of the virus. In the presence of an antiviral compound, the cytopathogenicity is inhibited and the fluorescent signal maintained. Stock solutions of the various compounds in DMSO (10 mM) were prepared. On day -1, the test compounds were serially diluted in assay medium (DMEM supplemented with 2% *v/v* FCS). The plates were incubated (37 °C, 5% CO_2_ and 95% relative humidity) overnight. On day 0, the diluted compounds were then mixed with SARS-CoV-2 at 20 TCID50/well and VeroE6-eGFP cells corresponding to a final density of 25,000 cells/well in 96-well blackview plates (Greiner Bio-One, Vilvoorde, Belgium; Catalog 655090). The plates were incubated in a humidified incubator at 37 °C and 5% CO_2_. At 4 days p.i., the wells were examined for eGFP expression using an argon laser-scanning microscope. The microscope settings were excitation at 488 nm and emission at 510 nm and the fluorescence images of the wells were converted into signal values. The results were expressed as EC_50_ values defined as the concentration of compound achieving 50% inhibition of the virus-reduced eGFP signals as compared to the untreated virus-infected control cells. Toxicity of compounds in the absence of virus was evaluated in a standard MTS-assay as reported previously [51].

The virus DENV3 clinical strain and A549 cells were used in the cell-based qRT-PCR antiviral assays. Evaluation of antiviral activity was performed as described previously [52]. Briefly, cells infected by dengue virus in the presence of 5-fold serial dilutions of compounds were maintained for 72 h. Supernatant was collected and the virus yield was measured by Cells Direct One-step qRT-PCR kit (Thermo Fisher), according to manufacturer’s instructions. Dose-response data were analyzed by nonlinear regression using GraphPad Prism software. The toxicity of compounds was evaluated in standard MTS analysis without virus in A549 cell line [51]. Graphpad Prism 9.2.0 was used to calculate EC_50_ (half maximal effective concentration) and CC_50_ (half cytotoxic concentration) values.

### 3.6. In Silico Study 

The protein structures for computational simulation were downloaded from the Protein Database Bank (PDB). SARS-CoV-2 RdRp/RNA complex (PDB ID: 7BV2) and Dengue Virus NS5 RNA Dependent RNA Polymerase Domain (PDB ID: 2J7U) was selected as docking receptors. 

Docking simulation procedure: all the calculation processes were supported by the corresponding modules of Schrodinger 2021-4 suite (www.schrodinger.com accessed on 13 November 2022) and was performed on DELL Precision T5500 workstation. Firstly, compounds were optimized with the Ligprep model with default parameters, and a pair of chiral isomers are generated for each compound. The ionic state under the physiological condition of ligand (pH = 7.0) was added; OPLS4 force filed was selected to optimize and obtain the ligand molecules required for screening. The preparation of the protein was completed by the protein preparation wizard module. A series of processes such as hydrogenation, charging, elimination of conflicting amino acid residues and energy minimization of the protein crystal structure were carried out with default parameters, and then the Remdesivir molecule of the composite crystal structure is extracted to obtain the receptor protein. The binding position of Remdesivir was used to locate the receptor grid for ligands. Finally, the Glide module [53,54] was used to dock the optimized ligands with the receptor protein with extra precision (XP). At the same time, baicalein and baicalin were used as training set to adjust molecular docking method, optimize parameter settings, and obtain a strong predictive docking model. According to the established docking model, Schrodinger 2021-4 Glide XP module was used for molecular docking of the compounds. The docking poses were visualized by Pymol (Schrödinger, LLC. DeLano Scientific, San Francisco, CA, USA, https://pymol.org accessed on 13 November 2022).

MD simulations were performed to further investigate the dynamic interactions between RdRp and the polyphenols. All simulations were conducted by using Schrodinger version 2021-4, and employed OPLS-4 force field. The previously generated polyphenol docking complex was employed as the starting coordinates, which was then filled into a proper box and solvated with water (TIP3P). The whole system was then added corresponding Na^+^ or Cl^-^ to neutralize all charges. Then 0.15 M NaCl was additionally added to simulate salt concentration under physiological condition. The whole system was relaxed with default set and the productive simulation was then performed for 500 ns under standard state (300 K, 1 bar). The result trajectories were then analyzed and the RMSD of the ligand was calculated.

## 4. Conclusions 

Through an overall screening of SARS-CoV-2 RdRp inhibition of our in-house compound library, 11 hit compounds were preliminarily selected. The hit compounds were verified through SARS-CoV-2 and DENV RdRp inhibition experiment. **DF-47** exhibited the IC_50_ = 9.2 ± 1.1 μM against SARS-CoV-2 RdRp, meanwhile **DF-51** inhibited RdRp function of DENV with an IC_50_ value of 4.8 ± 0.7 μM. Subsequently, the binding mode of ligands and RdRp was predicted through molecular docking, then validated by molecular dynamics simulation and free energy calculation. The results confirmed our hypothesis that the polyphenols chelate with two Mg^2+^ at the active site of RdRp, thereby inhibiting its replication function unlike the reported nucleoside RdRp inhibitors which cause chain-termination or lethal mutation. Regretfully, the in vitro activity at the cellular level was unsatisfying, which was attributed to low membrane permeability. Computational prediction suggested that **DF-47** and **DF-51** could be modified to their prodrugs to improve the cell activity. Meanwhile, for this series of polyphenols, antivirus activity screening towards a broader panel of RNA viruses should be conducted in the future.

In summary, a series of polyphenols that inhibit the activity of SARS-CoV-2 and DENV RdRp were discovered through library screening, target activity verification, cell activity determination and biophysical property prediction. The uncommon inhibition towards both coronavirus and flavivirus hints for their future development to broad-spectrum non-nucleoside polymerase inhibitors of RNA viruses.

## Figures and Tables

**Figure 1 molecules-28-00160-f001:**
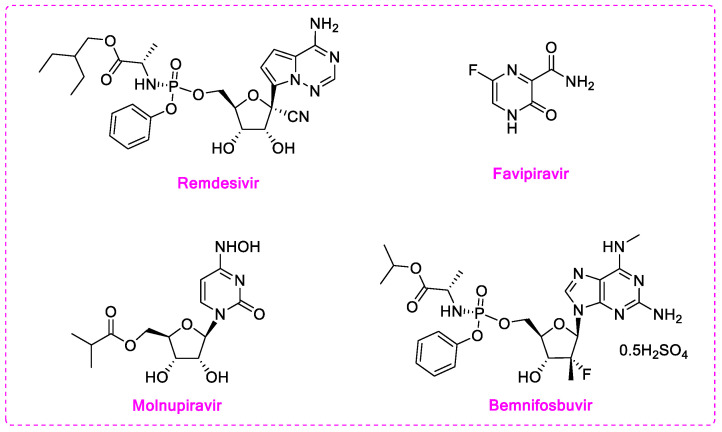
Structures of current nucleoside or nucleoside precursors RdRp inhibitors.

**Figure 2 molecules-28-00160-f002:**
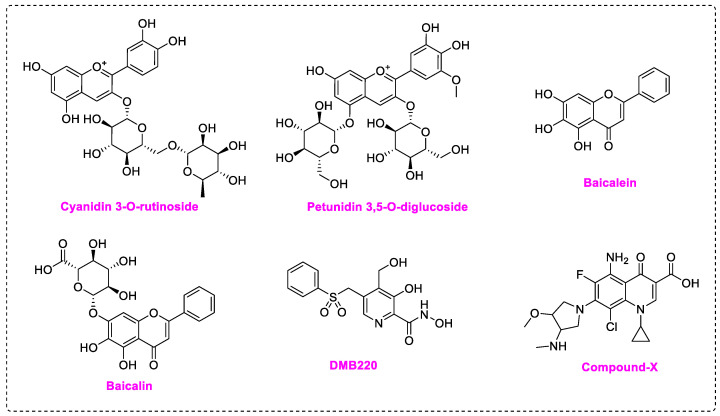
Structures of non-nucleoside inhibitors targeting SARS-CoV-2/DENV RdRp.

**Figure 3 molecules-28-00160-f003:**
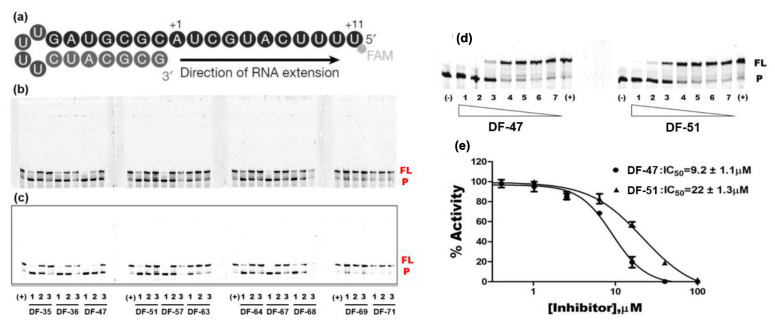
Inhibition of RNA-dependent RNA polymerase activity of 11 compounds. (**a**) Graphic representation of the 5′-6 FAM-labeled hairpin RNA substrate as primer-template used to monitor the inhibition of SARS-CoV-2 RdRp activity. +1 and +11, the positions of the first and the last nucleotide incorporated, respectively. (**b**) Gel picture with auto-contrast. Compound concentration: 1. 20; 2. 10; 3. 5 µM; (-): reaction in the absence of rNTPs; (+): reaction in the absence of compound. P: FAM-labeled RNA primer; FL: full-length RNA product. The data shown are from one representative experiment. (**c**) Gel picture with adjusted contrast. (**d**) Gel pictures of **DF-47** and **DF-51**. The concentrations of the compounds used are as follows: 1. 100; 2. 40; 3. 16; 4. 6.4; 5. 2.6; 6. 1; 7. 0.4 µM. (**e**) Dose–response curves of **DF-47** and **DF-51**.

**Figure 4 molecules-28-00160-f004:**
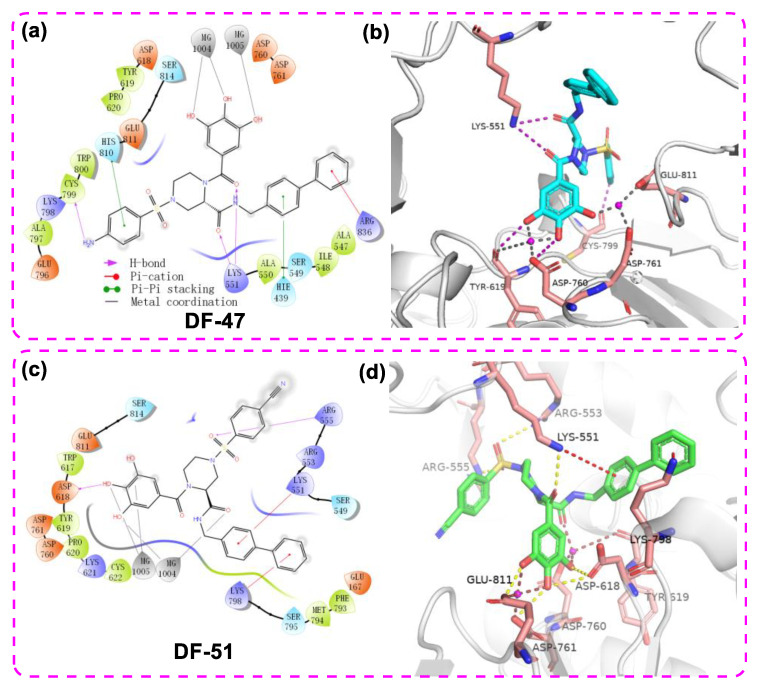
The 2D ligand–protein interaction diagram (**a**)/3D docking pose (**b**) of **DF-47** with DENV3 NS5 and the 2D ligand–protein interaction diagram (**c**)/3D docking pose (**d**) of **DF-51** with SARS-CoV-2 RdRp (PDB ID: 7BV2). The pink spheres represent Mg^2+^. The purple arrows indicate the hydrogen bonds; the red line represents Pi–cation interaction; the gray line represents metal coordination; the green lines represent pi-pi stacking.

**Figure 5 molecules-28-00160-f005:**
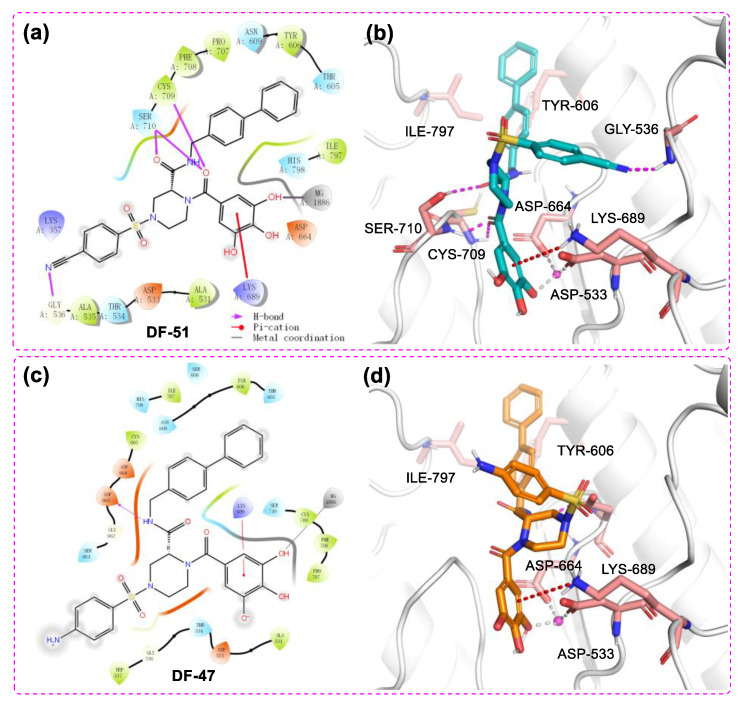
The 2D ligand–protein interaction diagram (**a**)/3D docking pose (**b**) of **DF-51** with DENV3 NS5 and the 2D ligand–protein interaction diagram (**c**)/3D docking pose (**d**) of **DF-47** with DENV3 NS5 RdRp domain (PDB ID: 2J7U). The pink spheres represent Mg^2+^. The purple arrows represent the hydrogen bonds; the red line represents Pi–cation interaction; the gray line represents metal coordination.

**Figure 6 molecules-28-00160-f006:**
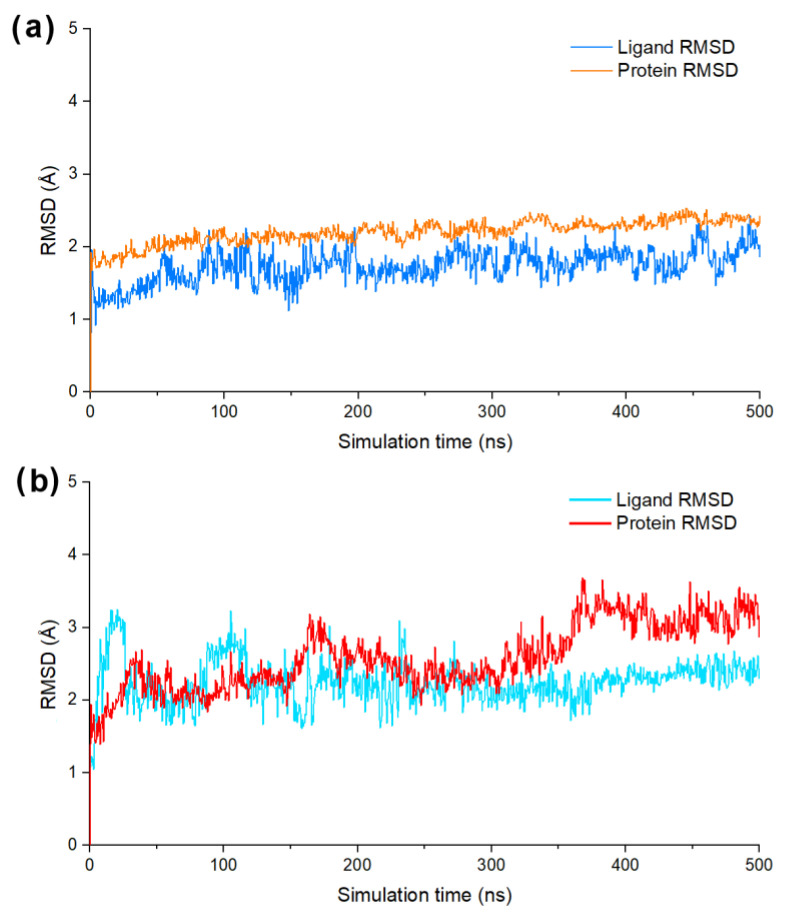
(**a**) The ligand/protein RMSD value of **DF-47** bind with SARS-CoV-2 RdRp; (**b**) the ligand/protein RMSD value of **DF-51** bind with DENV3 RdRp.

**Figure 7 molecules-28-00160-f007:**
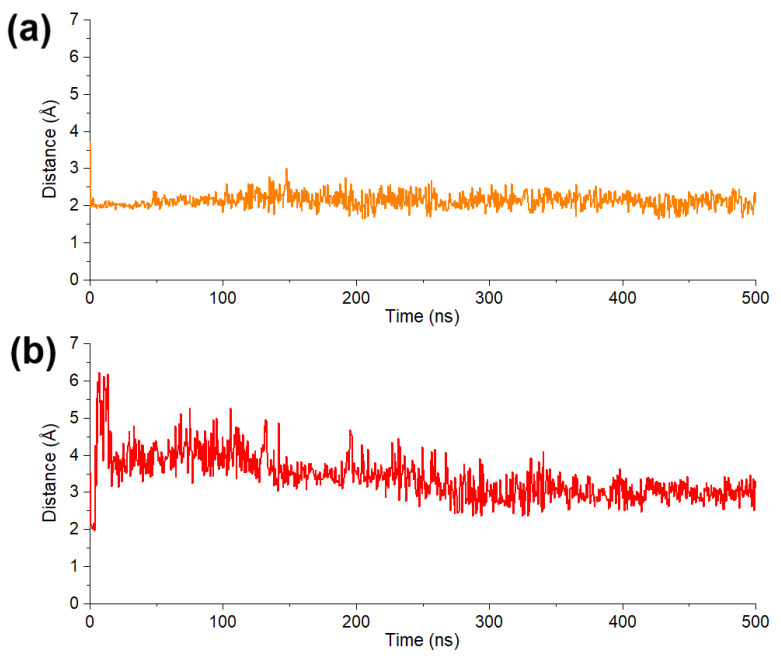
(**a**) Average distance between 4-OH of pyrogallol moiety in **DF-47** and two Mg^2+^. (**b**) Distance between 4-OH of pyrogallol moiety in **DF-51** and Mg^2+^.

**Table 1 molecules-28-00160-t001:**
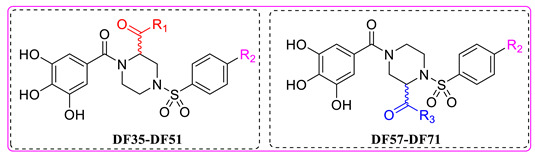
Compounds with top inhibition rates.

Compounds	R_1_	R_2_	Inhibition Rate (%)
**DF-35**	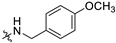	-NH_2_	81
**DF-36**	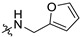	-NH_2_	87
**DF-47**	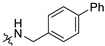	-NH_2_	92
**DF-51**	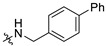	-CN	90
**DF-57**	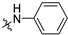	-NH_2_	85
**DF-63**	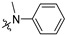	-NH_2_	75
**DF-64**	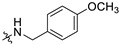	-NH_2_	80
**DF-67**	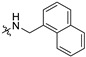	-NH_2_	83
**DF-68**	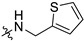	-NH_2_	73
**DF-69**	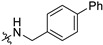	-NH_2_	72
**DF-71**	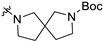	-NH_2_	70

**Table 2 molecules-28-00160-t002:** Inhibition of dengue virus RdRp activity assessed in a cell-free fluorescent plate assay.^a^.

Compounds	IC_50_ (µM)
**DF-47**	14 ± 2.3
**DF-51**	4.8 ± 0.7

^a^ IC_50_ (half-maximal inhibitory concentration) values were determined in RdRp assay using purified recombinant Dengue virus RdRp. Data are the average ± standard deviation of 2 independent experiments.

**Table 3 molecules-28-00160-t003:** Antiviral activity and cytotoxicity of polyphenols and positive controls.

Virus/Cells.	Compounds	EC_50_ (μM)	CC_50_ (μM)	Selective Index
SARS-CoV-2/ Vero E6	**DF-47**	>100	>100	-
	**DF-51**	>89	>100	-
	Remdesivir	0.046 ± 0.0002	10.1	219.6
	Baicalein	4.5	86	19.1
DENV3/A549	**DF-47**	51.9 ± 26.3	>100	>1.9
	**DF-51**	21.5 ± 14.9	>100	>4.6

**Table 4 molecules-28-00160-t004:** Permeability prediction value of selected compounds.

Compounds	Membrane dG Insert	Compounds	Membrane dG Insert
**DF-35**	−23.79	**DF-51**	−19.23
**DF-36**	−26.28	**DF-67**	−22.99
**DF-47**	−21.63	**DF-69**	−21.97
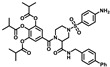 **DF-47-pro**	−5.50	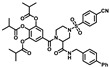 **DF-51-pro**	−5.42

## Data Availability

Data is contained within the article or Appendix A.

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
