# Peer review of "Identification of Polyphenol Derivatives as Novel SARS-CoV-2 and DENV Non-Nucleoside RdRp Inhibitors"

_molecules, 2022, doi:10.3390/molecules28010160_

Round 1

Reviewer 1 Report

This manuscript by Gao et al., describes the identification of two derivatives with inhibitory activity against SARS-CoV-2 and DFV RNA-dependent RNA-polymerases (RdRp), upon screening of an in-house compound library. These inhibitors, originally designed as potential HIV RNase H inhibitors, proved to possess quite interesting in vitro activity. The ‘drug repurposing’ approach for the discovery of novel scaffolds against human pathogens is often a very effective way in identifying new lead compounds. Although the new inhibitors did not prove to have in vivo efficacy, the authors provide a reasonable explanation for this limited activity. The manuscript is well written and provides all the necessary information concerning the biological evaluation of the newly identified inhibitors. Thus, it could be considered for publication in this journal, after minor revision.

Minor issues to be addressed:

- Page 2, line 59: Typically, Favipiravir is a pyrazinecarboxamide derivative and not a nucleoside. It requires in vivo conversion to its active form, the ribosylated triphosphate favipiravir, in order to act as an antiviral agent. This should be noted in the second paragraph of page 2. Also, in the legend of figure 1 (for example the title could be changed to: Structures of current nucleoside or nucleoside precursors RdRp inhibitors).

- Page 2, line 86: ‘petunidin 3-O-diglucoside’ should be corrected to ‘petunidin 3,5-O-diglucoside’.

- Pages 5 and 6, lines 218-221: The authors say that about 100 compounds of their in-house library were tested. These compounds were previously synthesized by their group and the corresponding literature is provided. According to the authors, these compounds were ‘polyphenols, 3H-pyrido[3,2-d]pyrimidin-4-one, and 4-hydroxyquinazoline analogues’. Upon examination of the structures in the provided literature, we see that all of these compounds are polyphenolic (galloyl) derivatives, with piperazine or piperidine side chains. Thus, no pyridopyrimidin-4-one or 4-hydroxyquinazoline compounds are described. If so, the authors should correct it in the manuscript.

- The authors should provide in the supplementary material a table, containing all the chemical structures of the compounds that were used in the initial screening.

- Page 9, figures 4 and 5: Pictures with higher resolution (especially for figure 5) should be provided.

- Supplementary material: The authors should add a paragraph with general information before the experimental section, where the instrumentation used should be described.

- Supplementary material: The authors describe in the experimental section the purity of the target derivatives (DF- products), but they do not provide any information about the method they used. If a HPLC method was used, they should provide the corresponding parameters (eluent system, flow rate, retention time for each compound, etc.). The corresponding chromatograms, at least for the two most active derivatives, should also be provided, as well.

Author Response

Thank you for your careful reading of our manuscript. The point-by-point response and revision details has been uploaded in the form of PDF file attachment.

Reviewer 2 Report

The manuscript entitled “Identification of Polyphenol Derivatives as Novel SARS-CoV-2 and DENV Non-Nucleoside RdRp Inhibitors” by Gao et al. is well written and organized. The authors identified two polyphenol derivatives DF-47 and DF-51 against SARS-CoV-2 and DENV by in silico study and experiment, which are potential to be developed as antiviral agents. Although, as for SARS-CoV-2, the in vitro activity at the cellular level was unsatisfying, the work provided some information for future studies. The following are some suggestions for the authors to strengthen their manuscript.

1. Why select RdRp of SARS-CoV-2 and DENV, not other viruses? High sequence homology, same protein structure, or something else, there should be addressed in detailed.

2. In Silico Study, its better list all the data associated with compounds (DF-47 and DF-51) and RdRp (SARS-CoV-2 and DENV3), from figure 4 to figure 6. For example, figure 4 should be included, DF-47 and SARS-CoV-2 RdRp; DF-47 and DENV3 RdRp; DF-51 and SARS-CoV-2 RdRp; DF-51 and DENV3 RdRp; and so on.

3. Line 148, .... an MOI of 0.001 TCID50/cell, the description maybe wrong. MOI and TCID50 are different, Either MOI, or TCID50, Please check.

4. Line 155-156, ...previously established SARS-CoV assay., add reference.

5. Line 181, ...EC50 and CC50 values, Whats CC50? Not mentioned in the text.

6. Line 280-281, In addition, DF-47 and DF-51 displayed no cytotoxicity for either Vero E6 or A549 cells.No data here, please add. 

Author Response

Thank you for your opinions and suggestions of our manuscript. The point-by-point response and revision details has been uploaded in the form of PDF file attachment.

Reviewer 3 Report

In this paper Gao et al., performed an in-house library screening to identify possible SARS-CoV-2 RdRp inhibitors. The authors identified DF-47 and DF-51 as inhibitors of SARS-CoV-2 and DENV using an RdRp enzymatic assay. However, they observed no antiviral activity in cell-based assays. Furthermore, the authors predicted very low compound permeability, which explain the absence of antiviral effect in cell-based assays. Finally, the authors describe, by in-silico analysis, that compounds DF-47 and DF-51 chelate Mg2+ at the active site of RdRp.

In general, the rational and logic of the manuscript is excellent. The manuscript guides the reader from enzymatic assays, in silico predictions for mechanism of action and prediction of permeability. However, the development and testing of permeable DF-47 and DF-51 derivatives is missing, reducing the impact of the manuscript.

The authors should consider this further work to strength the manuscript, however, this reviewer understands that this development could take a long time. So, one possibility to increase the impact of this paper is to reinforce the pan-RNA virus inhibition effect using enzymatic assays involving a set of RdRp complexes from other RNA viruses.

Author Response

Thank you for your positive comments of our manuscript. The point-by-point response and revision details has been uploaded in the form of PDF file attachment.

Round 2

Reviewer 3 Report

Thank you very much for your answers to the comments, I agree that in the future you will continue developing more permeable compounds and test them in a battery of RNA viruses.

Author Response

Response: Thank you for your affirmation of our manuscript and experimental work. As the manuscript needs no more revisions at this moment, we here keep it unchanged for publication in a few days.